# Intracellular expression of a fluorogenic DNA aptamer using retron Eco2

**Mahesh A Vibhute[†], Corbin Machatzke[†], Saskia Krümpel, Malte Dirks, Katrin Bigler, Daniel Summerer, Hannes Mutschler\***

Department of Chemistry and Chemical Biology, TU Dortmund University, Dortmund, Germany

## eLife Assessment

This study presents a method for expressing single-stranded DNA fluorescent aptamers in *E. coli* using a retron-based strategy. The evidence supporting the successful expression and folding of DNA aptamers is **solid**, with clear demonstration of fluorescence after extraction, though the aptamers do not function in living cells. The method represents an **important** technical advance that will likely become standard for DNA aptamer expression in bacterial systems.

**\*For correspondence:**
hannes.mutschler@tu-dortmund.de

[†]These authors contributed equally to this work

**Competing interest:** The authors declare that no competing interests exist.

**Abstract** DNA aptamers are short, single-stranded DNA molecules that bind specifically to a range of targets such as proteins, cells, and small molecules. Typically, they are utilized in the development of therapeutic agents, diagnostics, drug delivery systems, and biosensors. Although aptamers perform well in controlled extracellular environments, their intracellular use has been less explored due to challenges of expressing them in vivo. In this study, we employed the bacterial retron system Eco2 to express a DNA light-up aptamer in *Escherichia coli*. Our data confirms that structure-guided insertion of the aptamer domain into the non-coding region of the retron enables reverse transcription and biosynthesis of functional aptamer constructs in bacteria. The purified DNA aptamer synthesized under intracellular conditions shows comparable activity to a chemically synthesized control. Our findings demonstrate that retrons can be used to express short DNA aptamers within living cells, potentially broadening and optimizing their application in intracellular settings.

## Introduction

Aptamers are single-stranded nucleic acids that fold into distinct structures, allowing them to bind specifically to target ligands (*Dunn et al., 2017*). The ligand-binding sites of aptamers are tailored to the shape and charge of the ligand, enabling precise interactions. In particular, aptamers isolated through in vitro selection protocols serve as affinity reagents, positioning them as nucleic acid analogs of antibodies (*Keefe et al., 2010*; *Qian et al., 2022*). As such, they offer distinct advantages over their protein-based counterparts such as small size, as well as rapid and cost-effective manufacturing (*Qian et al., 2022*). An area of particular interest is the development of fluorescent light-up aptamers (FLAPs), (*Neubacher and Hennig, 2019*; *Zhou and Zhang, 2022*; *Ouellet, 2016*) which specifically bind conditionally fluorescent dyes and activate their fluorescence (*Chen et al., 2019*). The advent of FLAPs has also significantly advanced the study of folding of aptamers, as efficient folding can be determined as a function of fluorescence output. Pioneering work by Jaffery and coworkers has led to the proliferation of numerous FLAPs. Notable examples include the RNA-FLAPs Spinach (*Paige et al., 2011*), Broccoli (*Filonov et al., 2014*), Corn (*Song et al., 2017*), Mango (*Dolgosheina et al., 2014*), Squash (*Dey et al., 2022*), and Pepper (*Chen et al., 2019*). These fluorogenic RNA aptamers serve as

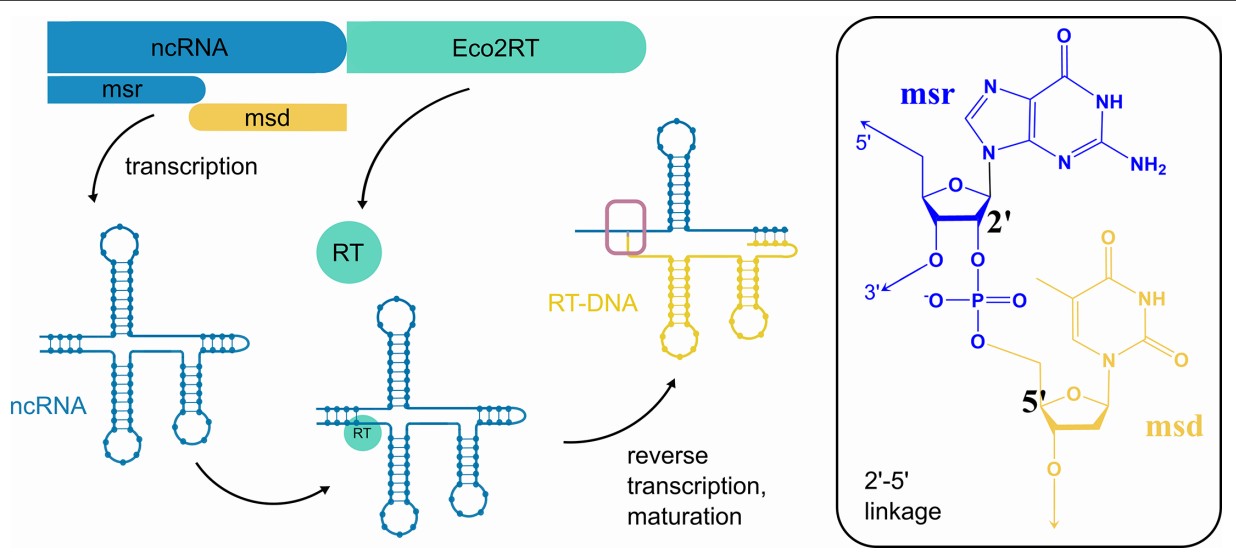

**Figure 1.** Eco2 retron locus and RT-DNA architecture. Schematic of the Eco2 gene locus, which encodes the non-coding RNA (ncRNA, blue) and the Eco2 reverse transcriptase gene (Eco2RT, green). The *msd* (yellow) region of the ncRNA is reverse transcribed into *msd* DNA, which is linked to the remaining *msr* via a 2'–5' linkage conserved guanosine close to the 3'-end of the *msr*, as indicated in the inset.

RNA mimics of fluorescent proteins and have transformed cellular imaging techniques to locate and understand RNA dynamics in vivo.

In contrast to RNA aptamers, DNA aptamers offer enhanced stability, greater ease of synthesis and chemical modification, as well as reduced cost, thereby facilitating more straightforward production and broader applicability in diverse fields, including therapeutics and diagnostics (*Qian et al., 2022*; *Zhu et al., 2015*). For example, Lettuce, a DNA FLAP that mimics the fluorescent properties of GFP, holds great promise for pathogen detection or as intracellular sensors for tumor cell recognition (*VarnBuhler et al., 2022*; *Passalacqua et al., 2023*; *Mou et al., 2023*). However, since most DNA aptamer studies focus on their development and characterization in vitro, their potential for applications inside living cells remains largely unexplored. A particular challenge in this context is the difficulty of biosynthesizing DNA aptamers in cells. Indeed, the ability to transcribe single-stranded RNA from DNA-based vectors has decisively contributed to the development of diverse RNA FLAPs for in-cell applications, such as the Broccoli aptamer (*Filonov et al., 2014*). Therefore, methods for the intracellular synthesis and functional studies of single-stranded DNA (ssDNA) aptamers are highly desirable. Only a few approaches for intracellular ssDNA synthesis have been reported, in all cases using exogenous systems such as phagemid vectors (*Lin et al., 2008*), phages (*Chen et al., 2004*), and eukaryotic retroviral reverse transcriptases (*Li et al., 2010*; *Chen et al., 2003*; *Chen, 2002*; *Elbaz et al., 2016*; *Alon et al., 2020*).

A more integrated, efficient, and potentially safer alternative to exogenous ssDNA expression systems could be the use of endogenous bacterial platforms. Retrons are prokaryotic genetic elements that are part of the bacterial phage defense machinery (*Palka et al., 2022*; *Wang et al., 2022*; *Millman et al., 2020*; *Bobonis et al., 2020*). Each retron typically consists of three components, namely a ncRNA transcript, an RT, and an effector protein. The ncRNA transcript adopts a specific secondary structure, nested between two repeat regions at its 5' and 3' ends that are complementary and hybridize (*Figure 1*). The RT recognizes the folded ncRNA and initiates reverse transcription from the 2'OH of a conserved guanosine, using the base-paired region as primer (*Simon et al., 2019*). The reverse transcription proceeds until about halfway along the ncRNA. Simultaneously, the RNA template is degraded by RNase H1, except for a stretch of approximately 5–10 RNA bases, which remain hybridized to the cDNA (*Palka et al., 2022*). This results in a hybrid RNA-DNA structure (RT-DNA), with the non-coding RNA and DNA elements *msr* and *msd*, respectively. *msr* and *msd* are covalently attached by a 2'–5' linkage at the conserved guanosine and the base paired region close to the 5' end of *msd* (3' end of *msr*). This ability of retrons to synthesize abundant ssDNA in vivo has generated considerable interest in using them as an alternative to exogenously delivered

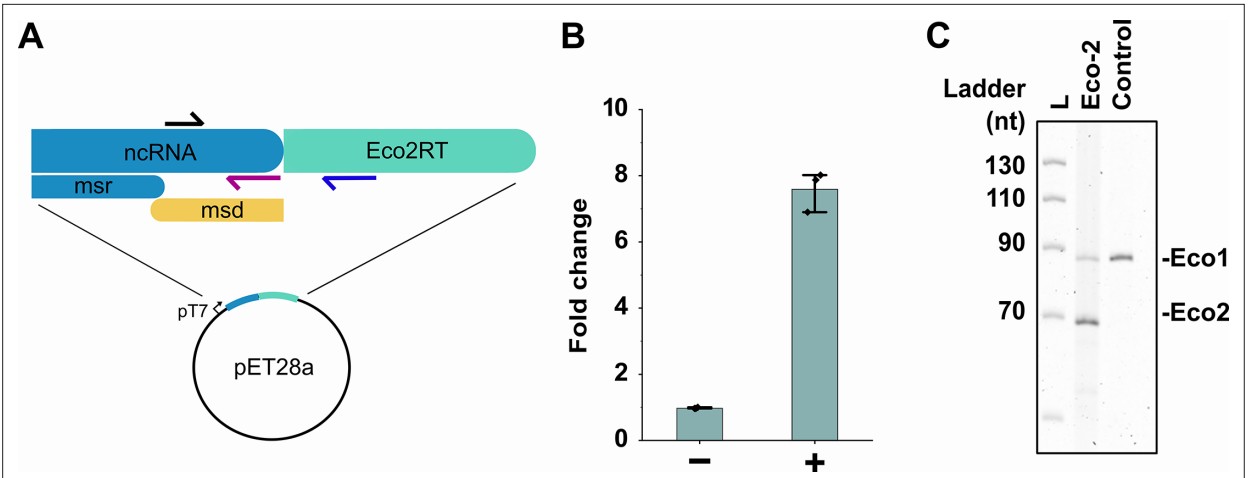

**Figure 2.** Recombinant expression of Eco2 RT-DNA. (**A**) Illustration of primer target sites for qPCR experiments to determine RT-DNA abundance. The black arrows indicate the forward primer that pairs either with the purple reverse primer (amplify both RT-DNA and plasmid DNA) or the blue reverse primer that only amplified the plasmid DNA. (**B**) Fold enrichment of the RT-DNA/plasmid template over the plasmid alone upon induction, as measured by qPCR; Unpaired $t$-test, induced versus uninduced: $p<0.0001$; n=3 biological replicates. (**C**) A TBE-Urea polyacrylamide gel, stained with SYBR Gold showing RT-DNA corresponding to retron Eco1 (90 nt) and retron Eco2 (70 nt).

The online version of this article includes the following source data and figure supplement(s) for figure 2:

**Source data 1.** Raw data corresponding to *Figure 2B*.

**Source data 2.** PDF file containing original PAGE gel indicating relevant bands, corresponding to *Figure 2C*.

**Source data 3.** Original image file of PAGE gel displayed in *Figure 2C*.

**Figure supplement 1.** Determination of half-life of RT-DNA.

**Figure supplement 1—source data 1.** Raw data corresponding to *Figure 2—figure supplement 1*.

DNA oligonucleotides in genome engineering and genome editing (*Simon et al., 2019*; *Simon et al., 2018*; *Ellington and Reisch, 2022*; *Schubert et al., 2021*; *Lopez et al., 2022*; *Lim et al., 2020*; *Sharon et al., 2018*; *Liu et al., 2023*).

While retrons have also been proposed as a potential method for expressing DNA aptamers (*Chen et al., 2003*; *Elbaz et al., 2016*), this approach has not yet been experimentally validated. Recent studies have indicated that retrons can indeed be leveraged to express functional ssDNA in vivo. For example, *Lopez et al., 2022* tested modifications to the ncRNA in case of retron Eco1 in order to boost RT-DNA abundance. More recently, *Liu et al., 2024*Liu et al. also used retron Eco1 to express 10–23 DNAzyme in vivo. However, it is poorly understood how the encoding of a functional ssDNA sequence in retrons will affect its activity. Such studies may lead to valuable design principles that would be especially important in the cellular context, as the RT-DNA is known to interact with not only the RT but, depending on the retron, also RNaseH and the retron associated effector proteins (*Palka et al., 2022*; *Wang et al., 2022*).

In this work, we sought to demonstrate expression of a DNA FLAP Lettuce as a model using retron Eco2 (*Lampson et al., 1989*). We first characterize expression of plasmid-borne wild type Eco2, determining abundance and intracellular stability of Eco2 RT-DNA. Subsequently, we test suitable integration sites of different length variants of Lettuce within the retron *msd* using in vitro proto-typing. After identifying effective insertion positions, we finally demonstrate fluorogenic activity of the expressed RT-DNA, embedded within the Lettuce aptamer. Our work demonstrates proof of concept of expressing DNA aptamers using retrons.

## Results

We first established an expression system for retron Eco2 by cloning the ncRNA and RT in a pET28a plasmid backbone, under the control of a T7 promoter. The plasmid was transformed in BL21AI cells that enable tight control of T7-based expression. The intracellular synthesis of Eco2 RT-DNA expression was induced with 1 mM IPTG and 0,2% arabinose ('Materials and methods') and confirmed by

denaturing PAGE of DNA extracts from induced cells (*Figure 2*). Using gel-based quantification of both RT-DNAs, we observed that Eco2 RT-DNA expression levels were approximately 3–4 times higher than those of genomically encoded Eco1 RT-DNA. To quantify expression levels, we further characterized Eco2 RT-DNA expression using qPCR (*Figure 2B*). Upon induction with IPTG and arabinose, we observed that the copy-number ratio between amplicon 1 (which can arise from amplification of both single-stranded RT-DNA and plasmid DNA, as shown in *Figure 2A*) and amplicon 2 (which can only be amplified from the plasmid construct) increased by almost eightfold in induced compared to uninduced cells. Assuming an average copy number of 15–20 for the pET28b expression, which utilizes a pBR322 origin (*Jahn et al., 2016*), we estimate that 100–200 single-stranded RT-DNA molecules are present in induced cells. With an average cytoplasmic volume of $7 \times 10^{-16}$ L of an *Escherichia coli* cell (*Phillips et al., 2012*), this corresponds to an intracellular concentration of approximately 250–500 nM.

Because no data are available on the stability of single-stranded RT-DNA in cells, we next determined the half-life of Eco2 RT-DNA. To this end, induced cells were thoroughly washed to remove residual IPTG and arabinose, thereby preventing de novo expression of RT-DNA. The washed cells were then resuspended in fresh medium and allowed to continue growing. RT-DNA levels were quantified by qPCR at defined time points. The gradual decrease in RT-DNA abundance could thus be attributed to either active degradation (e.g., by intracellular nucleases) or dilution due to cell growth and division. Intriguingly, no evidence for active RT-DNA degradation was observed as the decrease in RT-DNA could be entirely explained by dilution (*Figure 2—figure supplement 1*), indicating exceptional intracellular stability.

Having established the inducible expression of plasmid-encoded Eco2 in *E. coli*, we next sought to explore its suitability as a scaffold to host a functional Lettuce aptamer. Expression of ssDNA using retrons is typically achieved by encoding a cargo gene in the *msd* region of the ncRNA. This approach has been successfully demonstrated for recombineering donors (*Lopez et al., 2022*), protein-binding DNA sequences (*Lee and Kim, 2023*), and DNAzymes (*Liu et al., 2024*). However, it was not clear as to what extent the folding of the cargo aptamer sequence would be affected due to the extensive secondary structure of the Eco2 *msr-msd* RNA-DNA hybrid.

We speculated that the position of the Lettuce aptamer domain in the *msd* region would significantly affect its ability to fold into the functional three-dimensional structure. To test this hypothesis, we used a structure-guided approach to insert different variants of the Lettuce aptamer as cargo into four different positions in the *msd* region of retron Eco2 (*Figure 3A, B*). To identify four suitable insertion sites in the *msd*, we used the minimal free energy structure predicted for the Eco2 wild type *msd* by RNAfold (*Lorenz et al., 2011*) with energy parameters for DNA as guide to identify single stranded and/or loop regions where an insertion of the Lettuce domain was expected to not interfere with the native fold of the remaining *msd* sequence. We omitted the *msr*-overlapping region in the input sequences as it is known to form a duplex with the RTDNA as well as the *msr* RNA, as to the best of our knowledge, there is no algorithm capable of simulating mixed RNA-DNA sequences that are connected via a 2'–5' bond. The positions v1 and v3 are in two different predicted loop regions. The insertions site v2 is located in a single stranded that connects two hairpin loops, whereas position v4 is directly upstream of the predicted base-paired region between the *msd* and *msr* sequences.

The P1 stem of the original lettuce aptamer can be truncated down to four base pairs without significantly affecting its fluorescence output upon DFHBI-1T binding (*VarnBuhler et al., 2022*). We made use of this property in order to minimize the impact of Lettuce insertion on RT-DNA maturation while maintaining the ligand binding activity of Lettuce. Initially, we probed if truncated aptamer variants with P1 stem lengths of 4, 8, and 11 base pairs (designated 4L, 8L, and 11L) along with the full-length aptamer (FL) can still fold after embedding them into the integration sites of the emulated Eco2 RT-DNA.

Using chemically synthesized RT-DNA-Lettuce fusion constructs as prototypes, we first tested the fluorogenic potential of 4L after its insertion into the above-mentioned positions v1 to v4 of the Eco2 RT-DNA (resulting in the constructs 4LEv1 – 4LEv4) using in-gel staining with DFHBI-1T (*Figure 3—figure supplement 1*). We observed only very low levels of in-gel fluorescence compared to a standard full-length Lettuce (FL) or 4L alone, indicating that 4L folding was hampered upon embedding in the retron *msd* scaffold. In contrast, we observed a pronounced fluorescence signal for 8L when it was inserted position v4 (*Figure 3C*), demonstrating that this insertion site was compatible with Lettuce

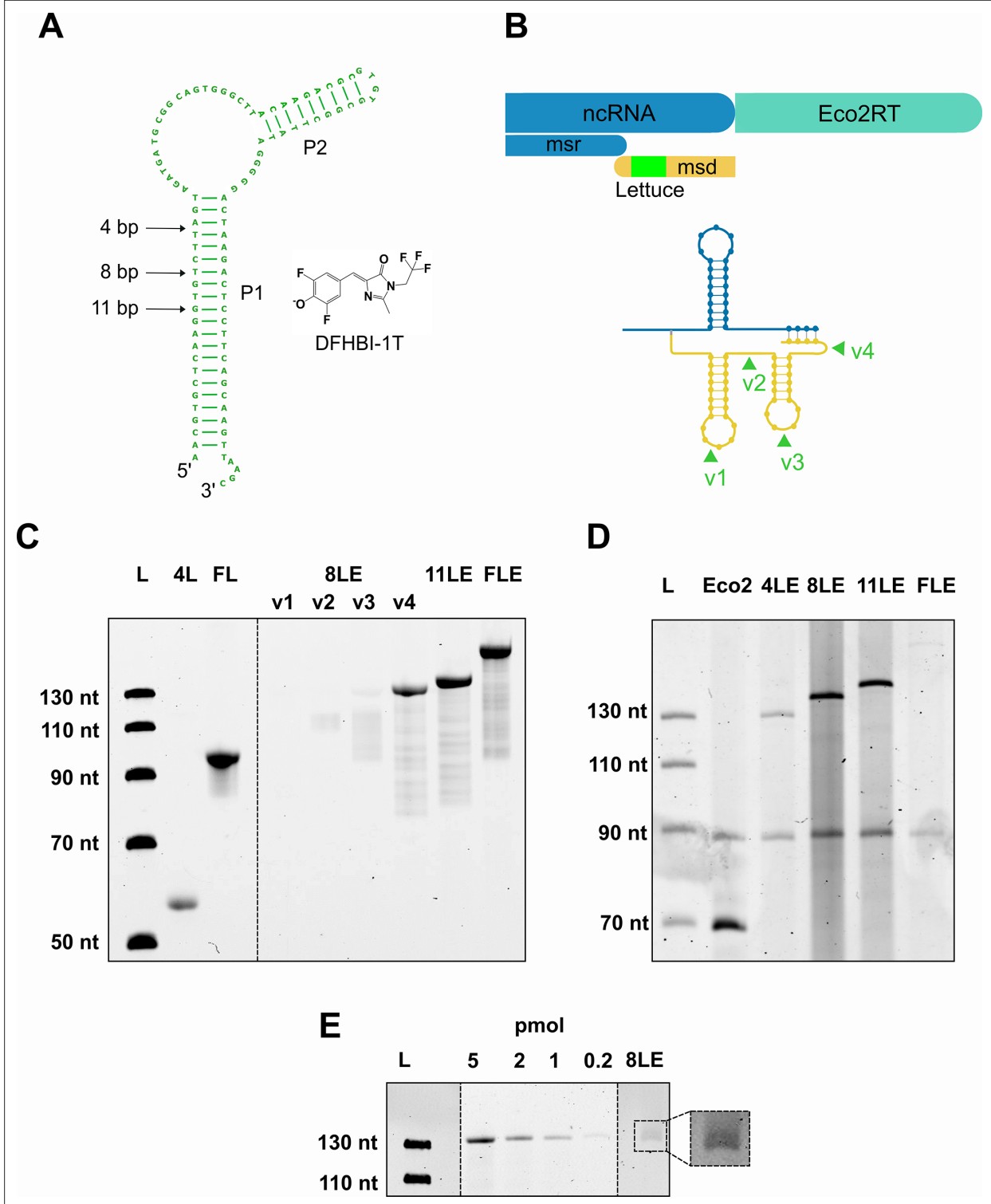

**Figure 3.** Expression of Lettuce-Eco2 fusion constructs in *E. coli* cells. (**A**) Schematic of Lettuce aptamer. The arrows indicate the length to which the P1 stem can be shortened without significant loss of FLAP functionality. (**B**) Schematic of insertion of Lettuce aptamer sequence in 4 distinct positions in the *msd* region of retron Eco2. The ssDNA structure was simulated using Vienna RNA fold with DNA parameters. Lettuce aptamer sequence was inserted at single-stranded and/or loop regions to minimize interference with the native fold of the *msd* sequence. (**C**) DFHBI-1T stained TBE urea-PAGE showing fluorescence of different oligonucleotides mimicking the Lettuce-Eco2 (LE) fusion constructs with varying P1 lengths: 8LE v1-v4 (134 nt), 11LE (140 nt), and FLE (166 nt). Free Lettuce with either the 4 nt P1 stem (4 L) or full-length P1 (FL) served as positive controls. (**D**) SYBR stained TBE urea-PAGE of extracted Eco2 wild type and Eco2-Lettuce RT-DNA fusions after expression in *E. coli*. In all variants, the lettuce aptamer was inserted into position v4

*Figure 3 continued on next page*

*Figure 3 continued*

of the *msd* scaffold. (**E**) DFHBI-1T stained TBE urea-PAGE showing fluorescence of 8LEv4 RT-DNA purified from *E. coli* cells. Lanes with different loading amounts in pmol of the synthetic RT-DNA standards are shown as comparison.

The online version of this article includes the following source data, source code, and figure supplement(s) for figure 3:

**Source data 1.** PDF file containing original PAGE gel indicating relevant bands, corresponding to *Figure 3C*.

**Source data 2.** Original image file of PAGE gel displayed in *Figure 3C*.

**Source data 3.** PDF file containing original PAGE gel indicating relevant bands, corresponding to *Figure 3D*.

**Source data 4.** Original image file of PAGE gel displayed in *Figure 3D*.

**Source data 5.** PDF file containing original PAGE gel indicating relevant bands, corresponding to *Figure 3E*.

**Source data 6.** Original image file of PAGE gel displayed in *Figure 3E*.

**Figure supplement 1.** DFBI-1T staining of 4LE oligonucleotides.

**Figure supplement 1—source data 1.** PDF file containing original PAGE gel indicating relevant bands, corresponding to *Figure 3—figure supplement 1B*.

**Figure supplement 1—source data 2.** Original image file of PAGE gel displayed in *Figure 3—figure supplement 1B*.

**Figure supplement 2.** Fluorescence titration experiments using chemically synthesized Lettuce-Eco2 v4 variants.

**Figure supplement 2—source data 1.** Raw data corresponding to *Figure 3—figure supplement 2*.

**Figure supplement 2—source code 1.** Source code used for fitting data from *Figure 3—figure supplement 2*.

**Figure supplement 3.** Detection of extracted RT-DNA constructs.

**Figure supplement 3—source data 1.** PDF file containing original PAGE gel indicating relevant bands, corresponding to *Figure 3—figure supplement 3*.

**Figure supplement 3—source data 2.** Original image file of PAGE gel displayed in *Figure 3—figure supplement 3*.

**Figure supplement 4.** Bulk in vivo fluorescence measurements of 8LEv1-4 with DFHBI-1T.

**Figure supplement 4—source data 1.** Raw data corresponding to *Figure 3—figure supplement 4*.

**Figure supplement 5.** Comparison of fluorogenic activity of Lettuce (DNA) and Broccoli (RNA) aptamers.

**Figure supplement 5—source data 1.** Raw data corresponding to *Figure 3—figure supplement 5*.

folding given a sufficient length of P1. Similarly, we observed strong in-gel fluorescence signals for 11LEv4 and FLEv4 comparable with native full-length Lettuce.

To cross-validate these results, we determined the dissociation constants between DFHBI-1T and the chemically synthesized Eco2-Lettuce v4 surrogate constructs using the reported Lettuce binding buffer (40 mM HEPES pH 7.5, 100 mM KCl, and 1 mM MgCl$_2$) (*VarnBuhler et al., 2022*). When increasing concentrations of each oligonucleotide were incubated in binding buffer containing a constant concentration of DFHBI-1T, we obtained well-defined fluorescence responses for all v4 constructs (*Figure 3—figure supplement 2*). When fitting the binding isotherms with a quadratic association curve, we obtained dissociation constants ($K_d$) for 8LE, 11LE, and FLE that were comparable to the unembedded full-length aptamer (FL) ~0.12 µM (0.01, 2.46, 95% CI). Binding was weaker for the synthesized ssDNA mimic 4LE-v4, with a $K_d$ of ~6 µM (4.92, 7.18, 95% CI), which was well in agreement with our in-gel fluorescence experiments.

Having confirmed that the v4 surrogate of 8LE, 11LE, and FLE constructs maintained the light-up aptamer properties of the embedded Lettuce sequences, we investigated whether the different length variants could be expressed in *E. coli* after integration into the plasmid-borne *msd* region of retron Eco2. PAGE analysis of RT-DNA from induced *E. coli* BL21-AI expressing the Eco2-Lettuce fusion constructs confirmed complete synthesis of the 8LE (134 nt) and 11LE (140 nt) species, and low-level production of FLE (166 nt), despite their *msd*-encoded RT-DNAs being more than twice as long as the wild-type (67 nt). Overall, the insertion of the Lettuce aptamer domain into the *msd* region of retron Eco2 resulted in a decrease in RT-DNA level compared to wild-type Eco2: While the length-corrected intensity of wild-type Eco2 was threefold higher than that of the endogenous Eco1 band, the 4LE, 8LE, 11LE, and FLE constructs showed between ~25% (FLE) and ~65–90% (4LE, 8LE, and 11LE) of the length-corrected relative band intensities (*Figure 3D*).

Next, we tested whether heterologously expressed 8LEv4 RT-DNA retains its fluorogenic properties, as it remained unknown whether retron-derived aptamer-DNAs produced under intracellular conditions are functionally equivalent to chemically synthesized ssDNA oligonucleotides. Because

column-based purification could not be scaled efficiently due to the large amounts of plasmid DNA suppressing column binding of the short ssDNAs, we developed a modified TRIzol-based extraction protocol optimized for ssDNA isolation while minimizing co-purification of plasmid DNA. Although the new method yielded considerably cleaner RT-DNA preparations, overall recovery was lower, necessitating the use of larger culture volumes to obtain sufficient material for analysis. The new extraction method was evaluated in pilot experiments for the 8LEv1-4 variants, and PAGE analysis of the isolated RT-DNA from pilot-scale experiments confirmed full-length synthesis of all four position variants (*Figure 3—figure supplement 3*). This approach was then applied to larger volumes for preparative extractions. TBE-urea PAGE analysis with DFHBI-1T staining indeed revealed that the purified, intracellular ssDNA retains fluorogenic activity upon DFHBI-1T binding (*Figure 3E*) .

While induced cultures expressing 8LEv4 did not show measurable fluorescence above DFHBI-1T–independent, expression-related autofluorescence (*Figure 3—figure supplement 4*), the intracellular production of a fluorogen-binding RT-DNA supports the idea that retron-encoded ssDNA can, in principle, generate functional aptamers in cells. The limited in vivo signal likely reflects both the moderate RT-DNA abundance (≈100–200 molecules per cell) and the substantially lower fluorogenic efficiency of Lettuce compared with the RNA light-up aptamer Broccoli (~100-fold, *Figure 3—figure supplement 5*), which is optimized for in vivo imaging of RNA (*Filonov et al., 2014*). Nevertheless, the detection of fluorogenic activity in purified material establishes a proof-of-principle that retron-derived ssDNA can be used to display DNA aptamers inside living cells.

## Discussion

Bacterial retrons such as Eco2 provide a versatile platform for intracellular reverse transcription-based synthesis of single-stranded DNA. To our knowledge, our present study is the first to show that functional DNA aptamers that recognize small molecule ligands can also be expressed via retrons. Expanding the repertoire of retrons toward DNA aptamer expression promises to significantly expand the repertoire of functional non-coding DNAs and enable the creation of new sensors and regulators with a wide range of intracellular applications. At the same time, retron platforms provide a direct way to tailor DNA aptamers specifically to intracellular conditions. This is particularly important as our data show that both the low expression levels in combination with the moderate fluorogenic activity of Lettuce provide ample room for, for example, evolutionary improvement of retron-encoded DNA aptamer performance under intracellular conditions. We observed a significant decrease in Eco2 expression levels after integration of the Lettuce domain, suggesting that reverse transcription is hindered by the changes in the *msd* sequence (*Lopez et al., 2022*).

In Eco2, the RT is directly fused to a topoisomerase-primase-like (Toprim) effector domain, whose activity causes abortive infection (*Wang et al., 2025*; *Pausch et al., 2025*). Recent structural studies elucidated the structure of the Eco2 nucleoprotein complex and revealed that each RT-Toprim fusion protein forms a 1:1 complex with the 2′–5′ linked DNA-RNA hybrid that further assembles into a trimer (*Wang et al., 2025*). Interestingly, while the *msr* portion of each complex bridges adjacent RT–Toprim monomers and is integral to the structural integrity of the complex, the electron density of the *msd* beyond the stem downstream of the branched 2′–5′ linkage remained largely undefined, suggesting a strong degree of flexibility. The proximal *msd* stem is proposed to sterically block the nuclease active site of the Toprim effector domain; upon activation by the phage-encoded ssDNA nuclease DenB from T4-related phages, this inhibition is relieved, enabling Toprim to cleave tRNA and thereby inducing translational shutdown (*Wang et al., 2025*; *Pausch et al., 2025*). The electron density of the distal *msd* region was largely unresolved in the reported cryo-EM structure, indicating that it is not an integral structural element of Eco2 and likely tolerates mutations or insertions without preventing trimerization. In our study, we indeed observed no major growth defects upon insertion of the aptamer domain at positions v1 to v4, confirming the high degree of tolerance of the *msd* region distal to the 2′–5′ branch point. This suggests that alternative functional sequences could be similarly integrated into the distal *msd* segment, provided that flanking *msr* and *msd* regions do not disrupt folding. A major advantage of retron-based intracellular synthesis of ssDNA in vivo is its simplicity. The desired sequence can be cloned into the *msd* region of plasmid-borne or genomically encoded retron Eco2. The copy number of the aptamer can also, in principle, be tuned by inducer concentration, promoter strength, and plasmid copy number. An ever-increasing number of DNA aptamers have been generated against a wide range of targets, from small molecules, macromolecules to whole

cells (*Qian et al., 2022*; *Sharma et al., 2017*). The retron system used in this work could enable the exploration of cellular applications for this growing repertoire of DNA aptamers. A further advantage of expressing DNA aptamers with retrons is the exceptional intracellular stability of the *msd*-derived ssDNA in the retron complex (*Figure 2—figure supplement 1*). Finally, expression of related retron systems has been demonstrated in yeast (*Miyata et al., 1992*) and mammalian cells (*Lopez et al., 2022*; *Mirochnitchenko et al., 1994*), thus potentially widening the range of in vivo applications of DNA aptamers.

In conclusion, retron-based aptamer expression platforms present a promising avenue for the development of novel DNA-based applications. Future research aimed at optimizing intracellular synthesis may elucidate the conditions under which larger single-stranded DNAs can be synthesized at higher concentrations without compromising reverse transcription and may also identify the key factors influencing the optimal in vivo folding of cargo sequences.

# Materials and methods

## Key resources table

| Reagent type (species) or resource | Designation | Source or reference | Identifiers | Additional information |
|---|---|---|---|---|
| Gene (*Escherichia coli*) | Ec67 (Eco2) | https://doi.org/10.1016/j.cell.2020.09.065 | Ec67 (pAA11) | *Supplementary file 1a* |
| Strain, strain background (*E. coli*) | BL21(AI) | Thermo Fisher Scientific | Cat. #: C607003 | Chemically competent cells |
| Chemical compound, drug | Ready-Lyse Lysozyme Solution | Lucigen, Biozym | Cat. #:R1804M | |
| Chemical compound, drug | TRIzol Reagent | Thermo Fisher Scientific | Cat. #:15596026 | |
| Commercial assay or kit | Luna Universal qPCR Master Mix | New England Biolabs | Cat. #:M3003L | |
| Commercial assay or kit | Monarch Spin Plasmid Miniprep Kit | New England Biolabs | Cat. #:T1110L | |
| Chemical compound, drug | DFHBI-1T | Lucerna | Cat. #:410-10mg | Fluorophore |
| Commercial assay or kit | ssDNA/RNA Clean & Concentrator | Zymo | Cat. #:D7011 | |
| Chemical compound, drug | SYBR Gold Nucleic Acid Gel Stain (10,000X Concentrate in DMSO) | Invitrogen | Cat. #:S11494 | |

## Constructs and strains

The retron Eco2 sequence (*msr-msd* and RT) was obtained from *Millman et al., 2020*, and purchased from IDT as gblock, which was inserted into a pET28a vector under control of T7 promoter and *lac* operator. Plasmid assembly was performed using the HiFi DNA Assembly (New England Biolabs) according to the supplier's instructions. The resulting plasmids were transformed into chemically competent *E. coli* Top 10 cells. Plasmid sequences were verified by colony PCR and Sanger sequencing (Microsynth AG). Verified plasmids were then transformed into chemically competent BL21AI cells for further experiments. These cells harbor endogenous retron Eco1, which serves as an internal control and a cassette encoding T7 polymerase driven by arabinose-inducible araBAD promoter. Lettuce variant sequences were cloned into retron Eco2 plasmid using Q5 Site-Directed Mutagenesis Kit (New England Biolabs). Primer sequences and annealing temperatures were generated using the NEBaseChanger tool. The sequences of Retron Eco2 (Ec67), oligonucleotides (Eco2-Lettuce constructs), and qPCR primers are listed in *Supplementary file 1a,b,c* respectively.

## RT-DNA extraction

### Miniprep method

5 ml LB cultures supplemented with 50 µg/mL kanamycin were inoculated from single colonies of BL21AI cells transformed with the respective pet28b *msd*-Lettuce constructs and grown at 37°C with shaking at 220 rpm for 90 min. Cultures were then induced with 1 mM IPTG and 0.2% arabinose and incubated overnight for expression. The $OD_{600}$ of the overnight cultures was measured, and an amount of culture corresponding to 2 mL at $OD_{600}$=1 was used for miniprep. RT-DNA was extracted using the Monarch Plasmid Miniprep Kit (New England Biolabs) and eluted in 40 µL Milli-Q water. The

eluate was treated with RNase Cocktail Enzyme Mix (Invitrogen) at 37°C for 30 min and subsequently purified using the ssDNA/RNA Clean & Concentrator kit (Zymo).

The purified RT-DNA was analyzed on a 10% TBE-Urea polyacrylamide gel with a 1x TBE running buffer. Gels were stained with SYBR Gold (Invitrogen) and imaged on Sapphire Biomolecular Imager (Azure Biosystems). Band intensities were normalized by length of RT-DNA. RT-DNA abundance of Eco2 and Lettuce variants was quantified by determining fold change compared to the Eco1 band. Band intensity analysis was performed with ImageJ (*Schindelin et al., 2012*).

## TRIzol-based extraction method

25 mL pre-cultures supplemented with 50 µg/mL kanamycin were inoculated from single colonies of BL21AI cells transformed with the respective pet28b *msd*-Lettuce constructs and grown at 37°C with shaking at 220 rpm until they reached an $OD_{600}$ of 0.5–0.8. These pre-cultures were used to inoculate 750 mL cultures, which were then grown at 37°C and 150 rpm for 2 h. The cultures were then induced with IPTG (1 mM) and arabinose (0.2%) and allowed to grow overnight (~16 h) until they reached an $OD_{600}$ 2.4–2.9 and harvested by centrifugation at 4°C and 6000 × *g* for 30 min. The cell pellets were resuspended in TES buffer and lysozyme (Ready-Lyse Lysozyme Solution, Biozym) was added according to the manufacturer's instructions (25 mL TES +4.5 µL 30,000 U/µL Lysozyme). Samples were incubated for 15 min at room temperature and then aliquoted into twelve 50 mL conical tubes. TRIzol (20 mL) was added to each tube, and samples were gently vortexed until clear. Chloroform (4 mL) was then added, and samples were vortexed until homogeneous, followed by incubation for 5 min at room temperature. Phase separation was achieved by centrifugation for 30 min at 4°C and 12,000 × *g*. The aqueous phase (10 mL) was transferred to a fresh tube, mixed with 10 mL isopropanol, vortexed, and incubated for 2 h at −20°C. Nucleic acids were pelleted by centrifugation at 30,000 × *g* and 4°C for 1 h. The supernatant was removed, and the pellet was washed twice with 10 mL of 70% ethanol, each time centrifuging for 5 min at 4°C and 10,000 × *g*. Pellets were air-dried for 15 min and resuspended in 1000 µL nuclease-free water.

## In-gel staining of the RT-DNA

Following the crude extraction of chimeric RT-DNA/RNA using the above-described TRIzol-based method, 650 µL of the total RNA/retron material (1000 µL) was incubated with 250 mM NaOH and 6.25 mM EDTA at 80°C for 1 h. After incubation, the pH was neutralized by the addition of 250 mM HCl, after which the now RNA-free, single-stranded RT-DNA was purified using the ssDNA/RNA Clean & Concentrator Kit (Zymo Research) following the manufacturer's instructions. Typically, a total of three columns was used for ssDNA isolation with an elution volume of 9 µL of nuclease-free $H_2O$. The eluate from column 1 was used to elute ssDNA from columns 2 and 3 to enhance the total RT-DNA concentration. The entire sample (approximately 7.5 µL) was mixed with 15 µL of loading dye (10 mM EDTA in formamide) and heated to 95°C for 3 min, followed by immediate incubation on ice for 5 min. The sample was loaded into a single well of a 10% TBE-urea PAGE gel and electrophoresed for 80 min at constant power (20 W). The gel was washed twice with $ddH_2O$ to remove residual urea and then stained for 30 min in 50 mL DFHBI-1T staining solution (40 mM HEPES, pH 7.5, 100 mM KCl, 1 mM $MgCl_2$, 10 µM DFHBI-1T). After a final 5 min wash in $ddH_2O$ to remove excess dye, the gel was imaged on a Sapphire Bioimager using 488/518 nm for DFHBI-1T fluorescence and 520/565 nm for the Cy3-tagged ladder.

## qPCR

qPCR experiments were performed using Techne Prime Pro 48 Real-Time qPCR system, according to comparative $C_T$ method, (*Livak and Schmittgen, 2001*; *Schmittgen and Livak, 2008*) as described by *Lopez et al., 2022*. For qPCR analysis, cell cultures grown for ~16 h were used. 25 µL of bacterial culture at an $OD_{600}$=1 was mixed with 25 µL water and incubated at 95°C for 5 min to lyse cells. 0.3 µL of this boiled culture was used as templates in 30 µL qPCR reactions using Luna Universal qPCR Master Mix (New England Biolabs). qPCR quantification of RT-DNA abundance was performed using a set of three primers. Two primers were complementary to the *msd* region of Retron Eco2, and the third was an outnest primer complementary to non-retron elements of the expression plasmid. Two templates were thus amplified: one corresponding to an RT-DNA sequence that could use both the RT-DNA and the plasmid as templates, and a second that could use only the plasmid-encoded Eco2 as

a template. The difference between cycle threshold ($\Delta C_T$) of the inside (RT-DNA) and outside (plasmid) primer sets was then calculated. This $\Delta C_T$ was then subtracted from the $\Delta C_T$ of uninduced samples. Fold enrichment of RT-DNA/plasmid over plasmid alone was then calculated as $2^{-\Delta\Delta CT}$, for each biological replicate. Presence of RT-DNA results in a fold change >1.

## Equilibrium binding measurements

Equilibrium binding measurements were performed on a BMG Labtech Clariostar plus plate reader. Oligonucleotides corresponding to different variants of Lettuce embedded in the retron scaffold were synthesized by IDT (*Supplementary file 1b*). Sample fluorescence was measured in a Greiner 384-well black-bottom plate with excitation at 470±8 nm and emission at 515±20 nm. The samples were prepared by mixing 40 mM HEPES pH 7.4, 100 mM KCl, 1 mM MgCl$_2$ with different concentrations of DNA. Samples were then heated to 90°C for 2 min, followed by incubating at room temperature for 5 min. After that 1 µM of DFHBI-1T was added to the mix. The samples were then briefly vortexed and spun down and pipetted in triplicate into a 384-well plate. Samples were then incubated for 60 min at RT in darkness, followed by fluorescence measurements on the microplate reader.

The datasets were fitted using *Equation 1*:

$$F = C_1 \cdot \frac{\left(A_t + B_t + k_D\right) - \sqrt{\left(A_t + B_t + k_D\right)^2 - 4 \cdot \left(A_t \cdot B_t\right)}}{2 \cdot B_t} + C_0 \tag{1}$$

where $F$ is the measured fluorescence, $A_t$ is the concentration of DNA, $B_t$ is the concentration of DFHBI-1T, and $C_0$ and $C_1$ are the lower and upper fluorescence plateaus. To account for uncertainty in the DFHBI-1T stock concentration, $B_t$ was also fitted under strong constraints. Curve fitting was performed using the least_squares function from the *scipy.optimize* package (https://docs.scipy.org/doc/scipy/reference/optimize.html) with shared global parameters $C_0$, $C_1$, and $B_t$ for all data series.

## Determination of half-life of Eco2 RT-DNA

Retron RT-DNA forms a phage surveillance complex with the associated RT and effector protein (*Rousset and Sorek, 2023*; *Gao et al., 2020*; *Carabias et al., 2024*; *Wang et al., 2024*). Owing to its unique 'closed' structure (*Wang et al., 2022*) (with the ends of *msr* and *msd* joined by a 2′–5′ linkage and a base-paired region) and its non-coding function, the intracellular stability of RT-DNA is of particular interest. To assess the stability of Eco2 RT-DNA, we determined its half-life by qPCR.

We first induced retron Eco2 RT-DNA expression in BL21AI cells with 1 mM IPTG and 0.2% arabinose overnight. On the following day, cells were pelleted and washed twice with LB medium to remove residual IPTG and arabinose. The washed cells were then resuspended and used to inoculate a fresh culture in LB medium without inducers at an initial OD$_{600}$ of 0.2. The culture was grown at 37°C, and aliquots were taken at defined timepoints. For each time point, a cell suspension with OD$_{600}$=0.1 was prepared and incubated at 95°C for 5 min. 1 µL of this boiled, diluted culture was used as input for a 20 µL qPCR reaction. qPCR experiments were performed as described above.

Assuming RT-DNA degradation would occur by active degradation mechanisms (nuclease-mediated degradation) and dilution (cell growth and division), we determined the rate of degradation by the following equation:

$$C\left(t\right) = 1 + A \cdot e^{-kt} \frac{OD_{600}\left(t_0\right)}{OD_{600}\left(t\right)} \tag{2}$$

where $k$ is the degradation rate constant and the ratio $\frac{OD_{600}\left(t_0\right)}{OD_{600}\left(t\right)}$ is the dilution factor accounting for dilution caused by cell division. $OD_{600}\left(t\right)$ was determined by fitting the OD$_{600}$ values to the following equation describing logistic growth (*Figure 2—figure supplement 1A*):

$$OD_{600}\left(t\right) = \frac{OD_{max}}{1 + e^{-r \cdot \left(t - t_0\right)}} \tag{3}$$

After substituting $OD_{600}\left(t\right)$ with the function in *Equation 3* we fitted the experimental data for the fold change of the RT-DNA to *Equation 2* (*Figure 2—figure supplement 1B*). The best fit (red line) was obtained with a rate constant $k$ converging towards zero, consistent with an Eco2 RT-DNA

half-life that is beyond the detection limit of our assay. To put our estimates into context, we note that typical RNA half-lives are on the order of minutes in growing *E. coli* cells (**Milo and Phillips, 2016**). We therefore refitted the data using fixed half-lives of 15 and 30 min. In both cases, the resulting best-fit curves deviated substantially from the experimental data, consistent with an RT-DNA half-life that is far longer than the *E. coli* doubling time under these optimal growth conditions. While we cannot exclude ongoing RT-DNA production due to promoter leakiness, we expect this effect to be minor, as RT-DNA expression requires both IPTG and arabinose, which were removed prior to inoculating the growth medium with the starter culture. Overall, our data support exceptional RT-DNA stability.

## Bulk live-cell fluorescence measurements

5 mL LB cultures supplemented with 50 µg/mL kanamycin were inoculated from single colonies of BL21AI cells transformed with the respective pet28b *msd*-Lettuce constructs and grown at 37°C with shaking at 220 rpm for 90 min. Cultures were induced with 1 mM IPTG and 0.2% arabinose, followed by growth for ~16 h. A culture volume corresponding to 5 mL at $OD_{600}=1$ was pelleted and washed twice with 1 mL 1× PBS. The pellet was then resuspended in 150 µL 1× PBS and 40 µM DFHBI-1T was added to the cell suspension, followed by incubation at room temperature for 20 min. The stained cells were then used for plate reader measurements (3 × 50 µL in a 96-well plate). Fluorescence was measured on a BMG Labtech Clariostar Plus plate reader (Ex/Em 470/520 nm). To correct for pipetting and cell-density variation, cultures were diluted in 1× PBS and $OD_{600}$ was determined after fluorescence acquisition. $OD_{600}$ values were measured on a NanoDrop using a 25-fold dilution and used to normalize the fluorescence signal.

## Acknowledgements

The authors thank Indrayani Phadtare and members of the Mutschler lab for fruitful discussions. HM acknowledges support by the European Research Council (ERC) under the Horizon 2020 research and innovation program (grant agreement ID: 802000, RiboLife).

## Additional information

### Funding

| Funder | Grant reference number | Author |
|---|---|---|
| European Research Council | 802000,RiboLife | Hannes Mutschler |

The funders had no role in study design, data collection and interpretation, or the decision to submit the work for publication.

### Author contributions

Mahesh A Vibhute, Conceptualization, Resources, Data curation, Formal analysis, Validation, Investigation, Visualization, Methodology, Writing – original draft, Writing – review and editing; Corbin Machatzke, Resources, Data curation, Software, Formal analysis, Validation, Investigation, Visualization, Methodology, Writing – review and editing; Saskia Krümpel, Resources, Investigation; Malte Dirks, Resources, Investigation, Methodology; Katrin Bigler, Resources, Methodology; Daniel Summerer, Conceptualization, Validation; Hannes Mutschler, Conceptualization, Resources, Data curation, Formal analysis, Supervision, Funding acquisition, Validation, Visualization, Methodology, Writing – original draft, Project administration, Writing – review and editing

### Author ORCIDs

Mahesh A Vibhute ⓘ https://orcid.org/0000-0002-5230-5290
Hannes Mutschler ⓘ https://orcid.org/0000-0001-8005-1657

Reviewer #1 (Public review): https://doi.org/10.7554/eLife.99554.3.sa1
Reviewer #2 (Public review): https://doi.org/10.7554/eLife.99554.3.sa2
Author response https://doi.org/10.7554/eLife.99554.3.sa3

## Additional files

### Supplementary files
Supplementary file 1. Nucleic acid sequences used in this work.

MDAR checklist

### Data availability
All data generated or analyzed during this study are included in the manuscript and supporting files; source data files have been provided for figures.

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
