## [Editor Report · eLife Assessment]

This study presents a method for expressing single-stranded DNA fluorescent aptamers in *E. coli* using a retron-based strategy. The evidence supporting the successful expression and folding of DNA aptamers is **solid**, with clear demonstration of fluorescence after extraction, though the aptamers do not function in living cells. The method represents an **important** technical advance that will likely become standard for DNA aptamer expression in bacterial systems.

---

## [Referee Report · Reviewer #1 (Public review)]

Summary:

The authors use an interesting expression system called a retron to express single-stranded DNA aptamers. Expressing DNA as a single-stranded sequence is very hard - DNA is naturally double stranded. However, the successful demonstration by the authors of expressing Lettuce, which is a fluorogenic DNA aptamer, allowed visual demonstration of both expression and folding, but only after extraction in cells, but not in vivo (possibly because of the low fluorescence of Lettuce, or perhaps more likely, some factor in cells preventing Lettuce fluorescence). This method will likely be the main method for expressing and testing DNA aptamers of all kinds, including fluorogenic aptamers like Lettuce and the future variants / alternatives.

Strengths:

This has an overall simplicity which will lead to ready adoption. I am very excited about this work. People will be able to express other fluorogenic aptamers or DNA aptamers tagged with Lettuce with this system.

Weaknesses:

Some things could be addressed/shown in more detail, e.g. half-lives of different types of DNA aptamers and ways to extend this to mammalian cells.

---

## [Referee Report · Reviewer #2 (Public review)]

Summary:

This manuscript explores a DNA fluorescent light up aptamer (FLAP) with the specific goal of comparing activity in vitro to that in bacterial cells. In order to achieve expression in bacteria, the authors devise an expression strategy based on retrons and test four different constructs with the aptamer inserted at different points in the retron scaffold.

The initial version of this manuscript made several claims about the fluorescence activity of the aptamers in cells, and the observed fluorescence signal has now been found to result from cellular auto-fluorescence. Thus, all data regarding the function of the aptamers in cells have been removed.

Negative data are important to the field, especially when it comes to research tools that may not work as many people think that they will. Thus, there would have been an opportunity here for the authors to dig into why the aptamers don't seem to work in cells.

In the absence of insight into the negative result, the manuscript is now essentially a method for producing aptamers in cells. If this is the main thrust, then it would be beneficial for the authors to clearly outline why this is superior to other approaches for synthesizing aptamers.

---

## [Author Response]

The following is the authors’ response to the original reviews

Comment to both reviewers:

We are very grateful for the thoughtful and constructive comments from both reviewers. During the revision, and in direct response to these comments, we performed additional control experiments for the cellular fluorescence measurements. These new data revealed that the weak increase in green fluorescence reported in our original submission does not depend on retron-expressed Lettuce RT-DNA or the DFHBI-1T fluorophore, but instead reflects stress-induced autofluorescence of *E. coli* (e.g. upon inducer and antibiotic treatment).

We also benchmarked the fluorogenic properties of Lettuce against the RNA FLAP Broccoli and found that Lettuce is ~100-fold less fluorogenic under optimal in vitro conditions. Consequently, with the currently available, in vitro- but not in vivo-optimized Lettuce variants, intracellular fluorescence cannot be reliably detected by microscopy or flow cytometry. We have therefore removed the original flow cytometry / and in-culture-fluorescence data and no longer claim detectable intracellular Lettuce fluorescence.

In the revised manuscript, we now directly demonstrate that retron-produced Lettuce RT-DNA can be purified from cells and remains functional ex vivo with a gel-based fluorophore-binding assays. Together, these data clarify the current limitations of DNA-based FLAPs for in vivo imaging, while still establishing retrons as a viable platform for intracellular production of functional DNA aptamers.

**Reviewer #1 (Public Review):**
Summary:The authors use an interesting expression system called a retron to express single-stranded DNA aptamers. Expressing DNA as a single-stranded sequence is very hard - DNA is naturally double-stranded. However, the successful demonstration by the authors of expressing Lettuce, which is a fluorogenic DNA aptamer, allowed visual demonstration of both expression and folding. This method will likely be the main method for expressing and testing DNA aptamers of all kinds, including fluorogenic aptamers like Lettuce and future variants/alternatives.Strengths:This has an overall simplicity which will lead to ready adoption. I am very excited about this work. People will be able to express other fluorogenic aptamers or DNA aptamers tagged with Lettuce with this system.

We thank the reviewer for their thoughtful assessment and appreciate their encouraging remarks.

Weaknesses:Several things are not addressed/shown:(1) How stable are these DNA in cells? Half-life?

We thank the reviewer for this insightful question.

Retron RT-DNA forms a phage surveillance complex with the associated RT and effector protein[1-4]. Moreover, considering the unique ‘closed’ structure of RT-DNA[5] (with the ends of *msr* and *msd* bound either by 2’-5’ linkage and base paired region) and its noncoding function, we hypothesized that the RT-DNA must be exceptionally stable. Nevertheless, we attempted to determine half-life of the RT-DNA using qPCR for Eco2 RT-DNA. To this end, we designed an assay where we would first induce RT-DNA expression, use the induced cells to start a fresh culture without the inducers. We would then take aliquots from this fresh culture at different timepoints and determine RT-DNA abundance by qPCR.

We induced RT-DNA expression of retron Eco2 in BL21AI cells as described in the Methods. After overnight induction, cells were washed to remove IPTG and arabinose, diluted to OD_600_ = 0.2 into fresh LB without inducers, and grown at 37°C. At the indicated time points, aliquots corresponding to OD_600_ = 0.1 were boiled (95°C, 5 min), and 1 µL of the lysate was used as template in 20 µL qPCR reactions (see revised Methods for details).

Assuming RT-DNA degradation would occur by active degradation mechanisms (nuclease-mediated degradation) and dilution (cell growth and division), we determined the rate of degradation by the following equation

\begin{document}$C(t)=1+A \cdot e^{-k t} \frac{O D_{600}\left(t_{0}\right)}{O D_{600}(t)}$\end{document}

where is the degradation rate constant and the ratio \begin{document}$\frac{O D_{600}\left(t_{0}\right)}{O D_{600}(t)}$\end{document} is the dilution factor which takes into account dilution by cell division. *OD_600_(t)* was determined by fitting the OD_600_ measurements by the following the equation describing logistic growth:

\begin{document}$O D_{600}(t)=\frac{O D_{\max }}{1+e^{-r \cdot\left(t-t_{0}\right)}}$\end{document}

Which yields the plots shown in Figure 2–figure supplement 1.

After substituting *OD_600_(t)* by the function in equation (2), we fit the experimental data for the fold-change of the RT-DNA to equation (1). Interestingly, the best fit (red) was obtained with a converging towards zero suggesting that the half-life of the RT-DNA is beyond the detection limit of our assay. To showcase typical half-lives of RNA, which are in the range of minutes in growing *E. coli* cells[6], we refitted the data using constant half-life of 15 and 30 minutes. In both cases, simulated curve deviated significantly from the experimental data further confirming that the half-life of the RT-DNA is probably orders of magnitude higher than the doubling time of *E. coli* under these optimal conditions. While we cannot exclude that the RT-DNA is still produced as a result of promotor leakiness, but we expect this effect to be low as the expression of RT-DNA in *E. coli* AI cells requires both the presence of IPGT and arabinose, which were thoroughly removed before inoculating the growth media with the starter culture. Overall, our data therefore argues for an exceptional stability of the RT-DNA in growing bacterial cells.

We have now included this new experimental data in the supplementary information.

(2) What concentration do they achieve in cells/copy numbers? This is important since it relates to the total fluorescence output and, if the aptamer is meant to bind a protein, it will reveal if the copy number is sufficient to stoichiometrically bind target proteins. Perhaps the gels could have standards with known amounts in order to get exact amounts of aptamer expression per cell?

The copy number of RT-DNA can be estimated based on the qPCR experiments. We use a pET28a plasmid, which is low-copy with typical copy number 15-20 per cell[7]. We determined the abundance of RT-DNA over plasmid/RT-DNA, upon induction, to be 8-fold, thereby indicating copy number of Eco2 RT-DNA to be roughly around 100-200. Assuming an average aqueous volume of *E. coli* of 1 femtoliter[6], the concentration of RT-DNA is ~250-500 nM. We have added this information to the revised version of the manuscript.

(3) Microscopic images of the fluorescent *E. coli* - why are these not shown (unless I missed them)? It would be good to see that cells are fluorescent rather than just showing flow sorting data.

In the original submission, we used flow cytometry as an orthogonal method to quantify the fluorescence output of intracellularly expressed Lettuce aptamer, anticipating that it would provide high-throughput, quantitative information on a large population of cells. During the revision, additional controls revealed that the weak increase in fluorescence we had previously attributed to Lettuce expression was in fact a stress-induced autofluorescence signal that occurred independently of retron RT-DNA and DFHBI-1T. We have therefore removed these data from the manuscript and no longer claim detectable intracellular Lettuce fluorescence.

To understand this limitation, we compared the in vitro fluorescence of Lettuce with that of the RNA FLAP Broccoli, which is commonly used for RNA live-cell imaging. Under optimal in vitro conditions, Lettuce shows ~100-fold lower fluorescence output than Broccoli (new Figure 3–figure supplement 5). Given this poor fluorogenicity and the low intracellular concentration of retron RT-DNA (now derived from the qPCR experiments), we conclude that the current Lettuce variants are below the detection threshold for in vivo imaging in our system. We now explicitly discuss this limitation and the need for further (in vivo) evolution of DNA-based FLAPs in the revised manuscript.

(4) I would appreciate a better Figure 1 to show all the intermediate steps in the RNA processing, the subsequent beginning of the RT step, and then the final production of the ssDNA. I did not understand all the processing steps that lead to the final product, and the role of the 2'OH.

We thank the referee for this comment. We have now made changes to Figure 1, showing the intermediate steps as well as a better illustration of the 2’-5’ linkage.

(5) I would like a better understanding or a protocol for choosing insertion sites into MSD for other aptamers - people will need simple instructions.

We appreciate the reviewer for bringing up this important point. We simulated the ssDNA structure using Vienna RNA fold with DNA parameters. Based on the resulting structure, we inserted Lettuce sequence in the single stranded and/or loop regions to minimise interference with the native msd fold. We have now included this information in the description of Figure 3.

(6) Can the gels be stained with DFHBI/other dyes to see the Lettuce as has been done for fluorogenic RNAs?

Yes. We have now included experiments where we performed in-gel staining with DFHBI-1T for both chemically synthesized Eco2-Lettuce surrogates as well as the heterologously expressed Eco2-Lettuce RT-DNA. We have added this data to the revised Figure 3 (panel C and E).

(7) Sometimes FLAPs are called fluorogenic RNA aptamers - it might be good to mention both terms initially since some people use fluorogenic aptamer as their search term.

We thank the referee for this useful suggestion. We have now included both terms in the introduction of the revised version.

(8) What E coli strains are compatible with this retron system?

Experimental and bioinformatic analysis have shown that retrons abundance varies drastically across different strains of *E. coli*[8-10]. For example, in an experimental investigation of 113 independent clinical isolates of *E. coli*, only 7 strains contained RT-DNA[8]. In our experiments, we have found that BL21AI strain is compatible with plasmid-borne Eco2. The fact that this strain has a native retron system (Eco1) allowed us to use it as internal standard. However, we were also able express Eco2 RT-DNA in conventional lab strains such as *E. coli* Top 10 (data not shown), indicating both ncRNA and the RT alone are sufficient for intracellular RT-DNA synthesis.

(9) What steps would be needed to use in mammalian cells?

We appreciate the reviewer’s thoughtful inquiry. Expression of retrons has been demonstrated in mammalian cells by Mirochnitchenko et al[11] and Lopez et al[12]. For example, Lopez et al demonstrate expression of retrons in mammalian cell lines using the Lipofectamine 3000 transfection protocol (Invitrogen) and a PiggyBac transposase system[12]. We also mention this in the discussion section of the revised manuscript. Expression of retron-encoded DNA aptamers in mammalian cells should be possible with these systems.

(10) Is the conjugated RNA stable and does it degrade to leave just the DNA aptamer?

We are grateful to the reviewer for their perceptive question. This usually depends on the specific retron system. For example, in case of certain retron systems such as retron Sen2, Eco4 and Eco7, the RNA is cleaved off, leaving behind just the ssDNA. In our case, with retron Eco2, the RNA remains stably bound to the ssDNA, thereby maintaining a stable hybrid RNA-DNA structure[10,13]. During the extraction of RT-DNA, the conjugated RNA is degraded during the RNase digestion step, and therefore is not visible in the gel images.

**Reviewer #2 (Public Review):**
Summary:This manuscript explores a DNA fluorescent light-up aptamer (FLAP) with the specific goal of comparing activity in vitro to that in bacterial cells. In order to achieve expression in bacteria, the authors devise an expression strategy based on retrons and test four different constructs with the aptamer inserted at different points in the retron scaffold. They only observe binding for one scaffold in vitro, but achieve fluorescence enhancement for all four scaffolds in bacterial cells. These results demonstrate that aptamer performance can be very different in these two contexts.Strengths:Given the importance of FLAPs for use in cellular imaging and the fact that these are typically evolved in vitro, understanding the difference in performance between a buffer and a cellular environment is an important research question.The return strategy utilized by the authors is thoughtful and well-described.The observation that some aptamers fail to show binding in vitro but do show enhancement in cells is interesting and surprising.

We appreciate the reviewer’s thorough assessment.

Weaknesses:This study hints toward an interesting observation, but would benefit from greater depth to more fully understand this phenomenon. Particularly challenging is that FLAP performance is measured in vitro by affinity and in cells by enhancement, and these may not be directly proportional. For example, it may be that some constructs have much lower affinity but a greater enhancement and this is the explanation for the seemingly different performance.

We thank the reviewer for this insightful comment. In response, we conducted a series of additional control experiments to better understand the apparent discrepancy between the in vitro and in vivo data. These experiments revealed that the previously reported increase in intracellular green fluorescence is independent of retron-expressed Lettuce RT-DNA and DFHBI-1T, and instead reflects stress-induced autofluorescence of *E. coli* upon inducer and antibiotic treatment. Our original negative controls (empty wild-type Eco2, uninduced cells in the presence of DFHBI-1T) were therefore not sufficient to rule out this effect.

As a consequence, we have removed the earlier FACS data from the manuscript and no longer claim detectable intracellular Lettuce fluorescence. The reviewer’s comment prompted us to re-examine the fluorogenicity of our constructs in vitro. We found that the 4Lev4 construct folds poorly and produces very low signal in in-gel staining assays with DFHBI-1T. In contrast, the 8LE variant (8-nt P1 stem at position v4) shows the highest fluorescence in these in-gel assays (new Figure 3C). Nevertheless, even this construct remains 100-fold less fluorogenic than the RNA-based FLAP Broccoli (new Figure 3–figure supplement 5), and we were unable to detect its intracellular fluorescence above background (new Figure 3–figure supplement 4).

To still directly demonstrate that retron-embedded Lettuce domains that are synthesized under intracellular conditions are functional, we modified our strategy in the revision and purified the expressed RT-DNA from *E. coli*, followed by in-gel staining with DFHBI-1T (new Figure 3E). Despite the challenge of obtaining sufficient amounts of ssDNA, this ex vivo approach clearly shows that the retron-produced Lettuce RT-DNA retains fluorogenic activity.

The authors only test enhancement at one concentration of fluorophore in cells (and this experimental detail is difficult to find and would be helpful to include in the figure legend). This limits the conclusions that can be drawn from the data and limits utility for other researchers aiming to use these constructs.

We appreciate this excellent suggestion. In the original experiments, the DFHBI-1T concentration in cells was chosen based on published conditions for live-cell imaging of the Broccoli RNA aptamer[14], which is substantially more fluorogenic than Lettuce. Motivated by the reviewer’s comment, we explored different fluorophore concentrations and additional controls to optimize the in vivo readout. These experiments showed that the weak intracellular fluorescence signal is dominated by stress-induced autofluorescence[15] (possibly due to the weaker antitoxin activity of the modified *msd*) and does not depend on the presence of Lettuce RT-DNA or DFHBI-1T.

Given the combination of low Lettuce fluorogenicity and low intracellular RT-DNA levels, we concluded that varying the fluorophore concentration alone does not provide a meaningful way to deconvolute these confounding factors in cells. Instead, we shifted our focus to a more direct assessment of Lettuce activity: we now demonstrate that retron-produced Lettuce RT-DNA can be purified from *E. coli* and retains fluorogenic activity in an in-gel staining assay with DFHBI-1T (new Figure 3E). We believe this revised strategy provides a clearer and more quantitative characterization of the system’s capabilities and limitations than the initial in vivo fluorescence measurements.

The FLAP that is used seems to have a relatively low fluorescence enhancement of only 2-3 fold in cells. It would be interesting to know if this is also the case in vitro. This is lower than typical FLAPs and it would be helpful for the authors to comment on what level of enhancement is needed for the FLAP to be of practical use for cellular imaging.

In the revised manuscript, we directly address this point by comparing the in vitro fluorescence of Lettuce (DNA) and Broccoli (RNA) under optimized buffer conditions. These experiments show that Broccoli is nearly two orders of magnitude more fluorogenic than Lettuce (new Figure 3-figure supplement 5). Thus, the low enhancement observed for Lettuce in cells is consistent with its intrinsically poor fluorogenicity in vitro.

Based on this comparison and on reported properties of RNA FLAPs such as Broccoli, we conclude that robust cellular imaging typically requires substantially higher fluorogenicity and dynamic range than currently provided by DNA-based Lettuce. In other words, under our conditions, Lettuce is close to or below the practical detection limit for in vivo imaging, whereas Broccoli performs well. We now explicitly state in the Discussion that further evolution and optimization of DNA FLAPs will be required to achieve fluorescence enhancements that are suitable for routine cellular imaging, and we position our work as a first demonstration that functional DNA aptamers can be produced in cells via retrons, while also delineating the current sensitivity limits.

**Recommendations for the authors:**

**Reviewer #1 (Recommendations For The Authors):**
Addgene accession numbers are not listed - how is this plasmid obtained?

The sequence was obtained from Millman et al[16], and ordered as gblock from IDT. The gblock was then cloned into a pET28a vector by Gibson assembly. We have now included this in the methods section.

**Reviewer #2 (Recommendations For The Authors):**
Page 2, line 40 - FLAPS should be FLAPs

We have corrected this typo in the revised version.

References

(1) Rousset, F. & Sorek, R. The evolutionary success of regulated cell death in bacterial immunity. Curr. Opin. Microbiol. 74, 102312; 10.1016/j.mib.2023.102312 (2023).

(2) Gao, L. et al. Diverse enzymatic activities mediate antiviral immunity in prokaryotes. Science 369, 1077–1084; 10.1126/science.aba0372 (2020).

(3) Carabias, A. et al. Retron-Eco1 assembles NAD+-hydrolyzing filaments that provide immunity against bacteriophages. Mol. Cell 84, 2185-2202.e12; 10.1016/j.molcel.2024.05.001 (2024).

(4) Wang, Y. et al. DNA methylation activates retron Ec86 filaments for antiphage defense. Cell Rep. 43, 114857; 10.1016/j.celrep.2024.114857 (2024).

(5) Wang, Y. et al. Cryo-EM structures of *Escherichia coli* Ec86 retron complexes reveal architecture and defence mechanism. Nat. Microbiol. 7, 1480–1489; 10.1038/s41564-022-01197-7 (2022).

(6) Milo, R. & Phillips, R. Cell biology by the numbers (Garland Science Taylor & Francis Group, New York NY, 2016).

(7) Sathiamoorthy, S. & Shin, J. A. Boundaries of the origin of replication: creation of a pET-28a-derived vector with p15A copy control allowing compatible coexistence with pET vectors. PLOS ONE 7, e47259; 10.1371/journal.pone.0047259 (2012).

(8) Sun, J. et al. Extensive diversity of branched-RNA-linked multicopy single-stranded DNAs in clinical strains of *Escherichia coli*. Proc. Natl. Acad. Sci. U. S. A. 86, 7208–7212; 10.1073/pnas.86.18.7208 (1989).

(9) Rice, S. A. & Lampson, B. C. Bacterial reverse transcriptase and msDNA. Virus Genes 11, 95–104; 10.1007/BF01728651 (1995).

(10) Simon, A. J., Ellington, A. D. & Finkelstein, I. J. Retrons and their applications in genome engineering. Nucleic Acids Res. 47, 11007–11019; 10.1093/nar/gkz865 (2019).

(11) Mirochnitchenko, O., Inouye, S. & Inouye, M. Production of single-stranded DNA in mammalian cells by means of a bacterial retron. J. Biol. Chem. 269, 2380–2383; 10.1016/S0021-9258(17)41956-9 (1994).

(12) Lopez, S. C., Crawford, K. D., Lear, S. K., Bhattarai-Kline, S. & Shipman, S. L. Precise genome editing across kingdoms of life using retron-derived DNA. Nat. Chem. Biol. 18, 199–206; 10.1038/s41589-021-00927-y (2022).

(13) Lampson, B. C. et al. Reverse transcriptase in a clinical strain of *Escherichia coli*: production of branched RNA-linked msDNA. Science 243, 1033–1038; 10.1126/science.2466332 (1989).

(14) Filonov, G. S., Moon, J. D., Svensen, N. & Jaffrey, S. R. Broccoli: rapid selection of an RNA mimic of green fluorescent protein by fluorescence-based selection and directed evolution. J. Am. Chem. Soc. 136, 16299–16308; 10.1021/ja508478x (2014).

(15) Renggli Sabine, Keck Wolfgang, Jenal Urs & Ritz Daniel. Role of Autofluorescence in Flow Cytometric Analysis of *Escherichia coli* Treated with Bactericidal Antibiotics. J. Bacteriol. 195, 4067–4073; 10.1128/jb.00393-13. (2013).

(16) Millman, A. et al. Bacterial Retrons Function In Anti-Phage Defense. Cell 183, 1551-1561.e12; 10.1016/j.cell.2020.09.065 (2020).